# A Privacy-Preserving Trajectory Publishing Method Based on Multi-Dimensional Sub-Trajectory Similarities

**DOI:** 10.3390/s23249652

**Published:** 2023-12-06

**Authors:** Hua Shen, Yu Wang, Mingwu Zhang

**Affiliations:** School of Computer Science, Hubei University of Technology, Wuhan 430068, China; cshshen@hbut.edu.cn (H.S.); 102111071@hbut.edu.cn (Y.W.)

**Keywords:** trajectory publishing, privacy preservation, trajectory privacy, *k*-anonymity

## Abstract

With the popularity of location services and the widespread use of trajectory data, trajectory privacy protection has become a popular research area. *k*-anonymity technology is a common method for achieving privacy-preserved trajectory publishing. When constructing virtual trajectories, most existing trajectory *k*-anonymity methods just consider point similarity, which results in a large dummy trajectory space. Suppose there are *n* similar point sets, each consisting of *m* points. The size of the space is then $m^{n}$. Furthermore, to choose suitable *k*− 1 dummy trajectories for a given real trajectory, these methods need to evaluate the similarity between each trajectory in the space and the real trajectory, leading to a large performance overhead. To address these challenges, this paper proposes a *k*-anonymity trajectory privacy protection method based on the similarity of sub-trajectories. This method not only considers the multidimensional similarity of points, but also synthetically considers the area between the historic sub-trajectories and the real sub-trajectories to more fully describe the similarity between sub-trajectories. By quantifying the area enclosed by sub-trajectories, we can more accurately capture the spatial relationship between trajectories. Finally, our approach generates k−1 dummy trajectories that are indistinguishable from real trajectories, effectively achieving *k*-anonymity for a given trajectory. Furthermore, our proposed method utilizes real historic sub-trajectories to generate dummy trajectories, making them more authentic and providing better privacy protection for real trajectories. In comparison to other frequently employed trajectory privacy protection methods, our method has a better privacy protection effect, higher data quality, and better performance.

## 1. Introduction

In the contemporary mobile internet era, individuals increasingly depend on mobile devices and applications for activities such as information access, social networking, and online shopping, all facilitated by the collection and utilization of location data. Furthermore, the advancements in big data, artificial intelligence, and related technologies enable the utilization of users’ location data for purposes like optimization of traffic flows [1] and advertising [2], as well as intelligent medical systems [3]. However, collecting location data also raises many security and privacy concerns. Unauthorized use and sharing of location data can result in privacy breaches [4,5,6]. For instance, individuals may exploit location data for tracking and monitoring purposes [7], including socializing [8], navigation, and travel [9]. Additionally, location data can help rescuers determine affected people’s location and movement trajectories in disaster rescue [10], making location data security crucial for public safety. Therefore, protecting personal location privacy has become an essential part of personal privacy protection.

Despite the emergence of numerous location privacy protection techniques, the effectiveness of current approaches faces challenges regarding the relentless technical advancements of attackers [11]. Consequently, researchers have been continuously exploring how to protect users’ location privacy more effectively.

Trajectory privacy protection techniques have gained significant attention in research due to their capacity to safeguard the privacy of users’ location and behavioral trajectories, becoming an integral component of location privacy protection strategies. There are three fundamental approaches to achieving trajectory privacy: (1) Fake Trajectory [4,12]. This method involves the addition of fabricated trajectory points [12] or the exchange of two location pairs that are close in time and space [4] to obfuscate the authentic trajectory data to preserve privacy. However, this method is susceptible to exploitation by attackers who can discern real from fake trajectory points by scrutinizing time and speed parameters. (2) Sensitive Location Suppression [13]. This approach safeguards user privacy by eliminating or concealing sensitive locations, such as home addresses and workplaces. Nevertheless, if an attacker can access additional information, such as the user’s frequently visited locations, the user’s actual location can still be deduced. (3) Trajectory *k*-Anonymity [7,14]. This approach is an extension of *k*-anonymity, in which the user’s trajectory is divided into multiple sub-trajectories, and k−1 other users’ trajectories are chosen for each sub-trajectory to obscure the current user’s trajectory.

During the process of *k*-anonymizing trajectories, researchers leverage the similarity of certain trajectory data to anonymize them effectively, preventing attackers from discerning the user’s real trajectory. The method described in [15] for publishing an anonymous trajectory is based on the generation of secure starting and ending points, the generation of candidate sets containing secure starting and ending points based on user habits, the generation of k−1 anonymous trajectories in both directions, and the correction of the accessibility of location points on each anonymous trajectory using access probabilities. Zhou and Wang [16] combined a fog computing and *k*-anonymity approach to protect trajectories, using fog computing to provide users with local storage and mobility and *k*-anonymity techniques to construct hidden regions for each trajectory snapshot based on time-dependent query probabilities and transfer probabilities to solve the problems of real-time trajectory privacy protection and offline trajectory data protection for continuous queries. Nevertheless, the trajectory *k*-anonymity technique possesses certain limitations. For instance, attackers can deduce the actual trajectories of users by discerning information disparities (e.g., time, space) among different users [17,18]. As shown in Figure 1, the attacker must distinguish between the user’s real trajectory (represented by a solid line) and the first fake trajectory (represented by a dotted line), which the attacker can easily distinguish because the two trajectories take different paths in the middle segment, which is the area through which the attacker distinguishes the user’s real trajectory. The attacker cannot distinguish between the real trajectory and the second fake trajectory (represented by a dashed line) because their routes are identical. However, because the real trajectory is generated between 9:00 and 10:00 and the second fake trajectory is generated between 12:00 and 13:00, the attacker can infer the real trajectory of the user using the time information of the trajectories. Thus, it can be seen that the method of selecting trajectories that meet the requirements of trajectory anonymity is the key to trajectory *k*-anonymity.

To address the shortcomings of trajectory *k*-anonymity, this study [19] aims to address the privacy leakage problem caused by the increasing amount of personal information sources with the popularization of mobile devices. To prevent the disclosure of user privacy in the published spatio-temporal trajectory dataset, proposing a trajectory *k*-anonymization model called KPDP, which is based on point density and partitioning. The model optimizes current anonymization methods in regards to trajectory collection, partitioning, preprocessing and trajectory clustering algorithms. It effectively prevents re-identification attacks and minimizes data utility loss in *k*-anonymized datasets. Zhang [20] proposed the DLIP algorithm, which constructs a dummy location set by randomly selecting probabilistic similarity offset locations under guaranteed semantic distinction, improved security of location privacy, avoiding background knowledge attacks, edge information attacks, and homogeneity attacks to some extent. Yang [21] generated similar trajectories in real-time by the angle and distance between real trajectory points during continuous query service, which can also prevent edge information attacks when protecting user trajectories. Guo [22] introduced a new query privacy algorithm for continuous querying of location-based services that consider users with similar directions, speeds, and the same transmission patterns for anonymization, thus protecting users’ privacy throughout the query cycle. The literature [23] introduces a method measuring semantic trajectory similarity across dimensions like weighted time and space. In contrast to conventional methods, it offers a comprehensive view of semantic relationships. Existing trajectory similarity approaches focus on two-dimensional raw trajectories, while literature proposed MSM considers and weights similarity across all dimensions. In summary, we can resist the aforementioned attack by taking into account the similarity of multiple perspectives (such as distance similarity, direction similarity, speed similarity, time similarity, and transmission similarity) between the produced k−1 trajectories and the real trajectory while producing the k−1 trajectories. However, we discover that the majority of the methods discussed above determine how similar two trajectories are by calculating the similarity between position points extracted from the two trajectories. The process of producing k−1 dummy trajectories for a given trajectory usually includes four steps: The first one is to extract the start point, end point, and some other points from the given trajectory. The second one is to generate a point set of size *k* for each extracted point in which every point is similar to the corresponding extracted point in multiple respects. The third is to produce a dummy trajectory by randomly choosing a point from the point set corresponding to the start point of the given trajectory, the point sets corresponding to the other extracted points, and the point set corresponding to the end point of the given trajectory in turn. The fourth is to determine k−1 dummy trajectories that are most similar to the given trajectory from the produced dummy trajectories. Suppose we extract *n* points from the given trajectory. We will have *n* point sets of size *k*. So, we will generate $k^{n}$ dummy trajectories. Then, we need to choose k−1 trajectories that are most similar to the given trajectory from the $k^{n}$ dummy trajectories, which typically results in significant computational overhead.

To address the limitations of trajectory *k*-anonymity with minimal computational overhead, this paper introduces a privacy-preserving trajectory publishing method (PP-TPS) that relies on multi-dimensional similarities among sub-trajectories. PP-TPS is a method for achieving trajectory *k*-anonymity. To minimize the computational burden associated with creating k−1 similar trajectories from a given trajectory, leveraging historic trajectory data, PP-TPS initially divides both the target trajectory and the historic data into sub-trajectories. Then, PP-TPS filters historic sub-trajectories for each sub-trajectory of the trajectory according to the computed multi-dimensional similarities between them. The main contributions of this paper are as follows:We design a *k*-anonymous privacy-preserving method (PP-TPS) based on sub-trajectories’ similarity. PP-TPS divides a given real trajectory in chronological order to obtain multiple segments (that is, sub-trajectories). Then, PP-TPS finds all sub-trajectories that are similar to these segments from historic trajectory sets. Finally, PP-TPS generates dummy trajectories for the given real trajectory by splicing similar sub-trajectories in chronological order. In other words, PP-TPS generates dummy trajectories by using real historic sub-trajectories to improve the authenticity of the generated dummy trajectories, enhancing privacy protection for real trajectories.PP-TPS considers the similarity of points between the historic trajectory and the real trajectory but also considers the similarity of the sub-trajectories between the historic and real trajectories. Moreover, PP-TPS proposes to measure the similarity of sub-trajectories by the area distance between them. This processing skill makes the processing process simpler and more efficient by drastically reducing the dummy trajectory space formed by simply taking the point similarity into account and skillfully avoiding solving the entire trajectory similarity.PP-TPS present a novel multidimensional similarity calculation approach that surpasses conventional similarity measures. It not only considers trajectory similarities in space but also factors in multiple dimensions. This cutting-edge method allows for a more comprehensive and precise depiction of trajectory relationships, offering a detailed and profound perspective for trajectory research.PP-TPS conducts comprehensive experiments on real datasets, and the results show that our method is more effective in terms of data availability and privacy protection compared to other methods.

The structure of the paper is outlined as follows: Section 2 provides an overview of related work. Section 3 introduces the relevant concepts and notations employed throughout the paper. Section 4 offers a comprehensive description of the proposed method, PP-TPS. Section 5 presents a comparative analysis of PP-TPS’s performance against other methods. Lastly, Section 6 summarizes the paper.

## 2. Related Work

The concept and basic principles of *k*-anonymity were first proposed by Latanya Sweeney [24]. *k*-anonymity technology aims to protect personal privacy by preventing an attacker from uniquely identifying a specific user. Gruteser and Grunwald [25] first applied *k*-anonymity methods to location services. They proposed a method to ensure users’ real locations cannot be accurately determined by temporal masking of the location information in time and space. Subsequently, *k*-anonymity techniques have gained significant traction in the domain of location-based services, with applications ranging from privacy-preserving traffic monitoring and management in traffic systems using *k*-anonymity technologies [26] to preventing the leakage of node location information in sensor networks through the adoption of *k*-anonymity technologies [27]. These techniques have also been instrumental in protecting user search history within search engines [28], securing the trajectories of devices [29], and preserving location privacy in mobile social networks [30]. This paper focuses on trajectory privacy by utilizing *k*-anonymization technologies. In summary, *k*-anonymization methods for trajectory privacy can be divided into two types: discrete point-based *k*-anonymization methods and trajectory-based *k*-anonymization methods.

### 2.1. Discrete Point-Based k-Anonymization

The discrete point-based approach [31,32,33] typically involves discretizing a trajectory into a sequence of points and then perturbing each point to attain *k*-anonymity, thereby enabling the creation of *k*-anonymized trajectories. The discrete point-based approaches are simple and intuitive. According to the dimension of similarity, this approach can be divided into two categories: discrete point-based *k*-anonymization with one-dimensional similarity and discrete point-based *k*-anonymization with multi-dimensional similarity.

The one-dimensional similarity *k*-anonymization method considers the similarity of locations from one perspective, such as spatial, semantic, or probabilistic, simplifying implementation. For example, the paper [34] proposed a new method called ARB to protect location privacy. The ARB method divides the space into grids and calculates the probability of queries within different grids based on historical data. Then, the query probabilities are used to generate anonymous regions for users to protect their location privacy. In [35], a location *k*-anonymity privacy protection scheme for vehicular networks was introduced. It involves dividing locations into multiple cells and assigning each cell a query probability based on the historical query distribution. By utilizing the Fréchet distance to measure trajectory similarity, it identifies the top 4k locations with the highest probabilistic similarity as candidates and delivers the optimal set of *k*-anonymous locations for straightforward and effective location privacy protection. In [29], the authors derived a gravity model describing user movement patterns from stored trajectory data. They established the flow relationships between locations, computed transfer probabilities among different grids based on the derived gravity model, and identified the virtual location with the highest location entropy to generate k−1 virtual locations, thereby preserving user trajectory data. Ref. [31] utilized a random walk to generate virtual location points within a specific geographic distance to ensure that all users in the steganography region have the same probability of being identified as real users. The scheme provided by [32] considered the user’s sensitivity to different location semantics and optimized collaborative segmentation to protect location privacy.

Although location points *k*-anonymity with similarity of one-dimensional location metric is easy to implement, an attacker can still threaten the user’s privacy with some side information. For example, an attacker may find that a user posts a post about a restaurant located in a certain area at one point in time, followed by a post about a movie theater located in a certain area at another point in time. By analyzing the temporal and geographical connections between these posts, the attacker can deduce the user’s unique trajectory based on the temporal references and location semantics (restaurant, movie theater), even when the user’s location has been anonymized. Researchers filtered them for multi-dimensional similarity to get virtual locations that are more similar to real locations. Literature [36] presents Enhanced-DLP, a novel, lightweight false location privacy protection scheme, aimed at tackling computational cost and additional information attacks challenges found in traditional schemes. Enhanced-DLP efficiently selects false locations to create *k*-anonymous sets using an improved greedy algorithm. The method incurs lower computational costs in selecting false locations and is capable of defending against attacks using supplementary information, in contrast with current approaches. The paper [37] considered the side information that the adversary may exploit. It first selected these virtual locations based on the entropy metric and subsequently introduced an improved DLS algorithm to maximize the dispersion of these selected virtual locations. Based on this, Yang et al. [38] took query probability and semantic location information as critical parameters, presented a virtual location selection model to evaluate the quality of virtual locations, and used a genetic algorithm-based optimization method to find the optimal solution, which ends up with a set of virtual locations with query probability as close as possible and also makes the locations in them as semantically and physically dispersed as possible. Unlike the above, the paper [39] built a semantic location tree (LST) and converted the semantic distance into the number of hops between nodes in the LST, designing a method that takes into account the semantic diversity and physical dispersion of locations, and combined the two objectives of geographic location and location semantics into a single objective optimization problem to improve efficiency, and ultimately produced an anonymous region. Concerning multi-dimensional location point similarity, Ref. [40] proposed a semantic-based method to protect trajectory anonymity. The method constructed a semantic region of sensitive location points by modeling each location point in the trajectory, where each sensitive point is associated with k−1 similar POI points. The method compared spatiotemporal and semantic similarities for anonymization to generate a usable anonymized trajectory dataset.

To select the k−1 dummy trajectories closest to the supplied trajectory, this approach must consider various virtual trajectories created by orderly combining location points of candidate point sets (anonymous point sets) corresponding to the selected locations on the given trajectory. This process often leads to considerable computational overhead.

### 2.2. Trajectory-Based k-Anonymization

The discrete-point *k*-anonymity-based approach significantly impacts trajectory accuracy and entails higher computational complexity, necessitating the processing of all individual points on the trajectory. In contrast, the trajectory *k*-anonymity-based approach treats a sequence of discrete points within the trajectory holistically to achieve *k*-anonymity, focusing on the trajectory’s overall characteristics for enhanced user privacy protection. This approach efficiently reduces computational overhead and enhances user privacy protection. Paper [41] presents a novel distributed *k*-anonymity algorithm to address the problem of location privacy leaks. The algorithm evaluates the similarity of users by analyzing their points of interest and social conduct, and picks users with a high similarity to form a group of anonymous collaborators. Ultimately, a homogenization algorithm guarantees the relative uniform distribution of the anonymized location points. Unlike the above scheme, ref. [42] primarily constrained the geographic location of trajectories and introduced a privacy-preserving method for measuring distance-based trajectory similarity. This method entailed constructing a trajectory network with spatial similarity serving as weights and selecting the node with the highest weight as the clustering center to create clusters meeting specific conditions. In the process of trajectory point perturbation, the trajectory points are shifted to the positions of neighboring nodes according to their degrees and specific conditions to achieve anonymization. This research [43] introduces sequential triple decision making and dynamic *k*-values to traditional *k*-anonymity to achieve personalized privacy protection. The proposed multi-level personalized *k*-anonymization model constructs a hierarchical decision table with attribute generalization trees and sensitive decision values supplied by the user. It utilizes dynamic *k*-value sequences to anonymize data. This study presents an extensive model for safeguarding privacy across multiple levels, thereby augmenting the use of sequential triple decision making in the privacy protection domain. Ref. [44] fused the similarity of multiple dimensions into trajectory similarity and proposed a personalized anonymization model that considers trajectory privacy and data utility. The model considers the effects of trajectory similarity, direction, and the trajectory distance on privacy and data utility and achieves a near-optimal set of *k*-anonymized trajectories by transforming the *k*-optimal trajectory selection problem into a constrained minimum spanning tree problem and using a trajectory graph model with a greedy strategy. In addition, the model pays full attention to users’ characteristics and requirements, exhibits strong scalability, and proves to be well suited to practical application scenarios. To further enhance the privacy protection effect of trajectories, this research [45] proposes the Privacy-Enhanced Distributed *k*-Anonymous Reward Mechanism (PEAK) to incentivize users to participate in distributed *k*-anonymity privacy protection in location services. PEAK does not need to trust third parties, establishes anonymous zones through currency transactions and location transmission, introduces role recognition and accountability mechanisms to limit malicious users, improves security and feasibility, and successfully establishes anonymous zones with a success rate of over 0.9, significantly reducing the utility of malicious users. Traditional *k*-anonymous trajectory privacy-preserving methods based on trajectory similarity usually use only one or two similarities between trajectories, resulting in ignoring the relationship between other attributes in the trajectories, and thus they cannot achieve the expected privacy-preserving effect.

For the above issues, the proposed PP-TPS method generates dummy trajectories by combining candidate sub-trajectories instead of candidate location points to reduce the computation cost and filters historic sub-trajectories according to multi-dimensional similarities (including time, space, semantics, direction, and probability) when creating candidate sub-trajectory sets to improve the intensity of privacy protection of user trajectories.

## 3. Preliminaries

### 3.1. Related Concepts

Trajectory

A trajectory is the path of an object moving through time and space. A trajectory usually consists of a series of coordinate points and a timestamp. A trajectory can be represented formally as *P* = $\{p_{1}$, *…*, $p_{n}\}$, and suppose we choose *n* location points of the trajectory to represent it. Here, $p_{1}$ and $p_{n}$ are the trajectory’s start and end, respectively. $p_{2}$, $p_{3}$, …$p_{n-1}$ are local extremum points of the trajectory. Moreover, $p_{i}$ = $(s_{i}$, $L_{i}$, $T_{i})$, $s_{i}$ = $(x_{i}$, $y_{i})$ is the latitude and longitude of the sample location point $p_{i}$; $L_{i}$ is $p_{i}$’s semantic information; and $T_{i}$ is its timestamp.

Sub-trajectory

A sub-trajectory is a series of consecutive location points intercepted from a trajectory. In this paper, we refer specifically to a series of two adjacent location points. For example, $st_{i}$ = $\{p_{i}$, $p_{i+1}\}$ is the *i*th sub-trajectory of the trajectory *P*, where *i* = 1, 2, …, n−1.

Local extremum point

A local extremum point is defined as the start point or end point of a sub-trajectory in a trajectory whose direction of motion is different from that of the adjacent sub-trajectories. $p_{i}$ is a local extremum point if it satisfies the following condition:(1)$$\begin{array}{c} |k_{\left(i-1,i\right)}-k_{\left(i,i+1\right)}\left|\;>\;\right|k_{\left(i-1,i\right)}|, \end{array}$$
where $k_{\left(i-1,i\right)}$ denotes the slope from $p_{i-1}$ to $p_{i}$.

### 3.2. Extracting Similar Location Points

PP-TPS filters similar location points out of all location points mainly based on their resemblance in terms of time, space, and semantics. If two points, denoted as $p_{i}$ and $p_{j}$, satisfy the requirements listed below, PP-TPS classifies them as similar.
(2)$$\left\{\begin{array}{c} Sim_{spa}\left(p_{i},p_{j}\right)>\delta_{spa}\\ Sim_{time}\left(p_{i},p_{j}\right)>\delta_{time}\\ Sim_{sem}\left(p_{i},p_{j}\right)>\delta_{sem} \end{array}\right.$$
where $Sim_{spa}\left(p_{i},p_{j}\right)$, $Sim_{time}\left(p_{i},p_{j}\right)$, $Sim_{sem}\left(p_{i},p_{j}\right)$ are the spatial similarity, the temporal similarity, and the semantic similarity between $p_{i}$ and $p_{j}$, respectively. In addition, $\delta_{time}$, $\delta_{spa}$, and $\delta_{sem}$ represent the thresholds related to spatial similarity, temporal similarity, and semantic similarity, respectively. These thresholds influence the selection of location points, ensuring that the filtered points exhibit significant similarities with the actual location points concerning spatial, temporal, and semantic attributes.

Below, we describe how to measure points’ spatial similarity, temporal similarity, and semantic similarity.

Spatial similarity [21]

Spatial similarity $Sim_{spa}\left(p_{i},p_{j}\right)$ refers to the degree of geographic closeness between two points. In our method, we calculate spatial similarity by employing the Euclidean distance:(3)$$\begin{array}{c} Sim_{spa}\left(p_{i},p_{j}\right)=1-\frac{d(p_{i},p_{j})}{1+d(p_{i},p_{j})}=\frac{1}{1+d(p_{i},p_{j})}, \end{array}$$
where $d(p_{i},p_{j})$ = $\sqrt{{\left(x_{i}-x_{j}\right)}^{2}+{\left(y_{i}-y_{j}\right)}^{2}}$ is the Euclidean distance between $p_{i}$ and $p_{j}$.

Temporal similarity

Temporal similarity $Sim_{time}\left(p_{i},p_{j}\right)$ refers to the degree of similarity between two points in time. The formula for temporal similarity is:(4)$$\begin{array}{c} Sim_{time}\left(p_{i},p_{j}\right)=\frac{1}{1+d_{time}\left(p_{i},p_{j}\right)}, \end{array}$$
where $d_{time}\left(p_{i},p_{j}\right)$ = $|T_{i}-T_{j}|$, $T_{i}$ (respectively, $T_{j}$) is the corresponding timestamp of the point $p_{i}$ (respectively, $p_{j}$).

Semantic similarity

Semantic similarity $Sim_{sem}\left(p_{i},p_{j}\right)$ refers to the similarity between two points in the semantic space. PP-TPS measures the semantic similarity between two points $p_{i}$ = $\{s_{i}$, $L_{i}$, $T_{i}\}$ and $p_{j}$ = $\{s_{j}$, $L_{j}$, $T_{j}\}$ based on WordNet.

WordNet [20,46] is an English vocabulary database containing a large number of English words and the relationships between them and their semantics. The vocabulary in WordNet can be categorized according to semantic classes, each of which contains words with the same or similar meanings. Words in WordNet may contain more than one semantic meaning, so it is necessary to consider multiple semantics when we calculate the location semantic similarity between $p_{i}$ and $p_{j}$.

Location semantics pertain to the semantic categorization of a location point, symbolized as *L*. Using geo-tags, these location semantics have been categorized into ten separate categories for the sake of this study. These categories include residential areas, commercial activities, leisure and entertainment, education and culture, and transportation. The acquisition of location semantics is contingent upon map information. The detailed information is shown in Table 1 [30].

If $L_{i}$ = $L_{j}$, the semantic distance $Sim_{sem}\left(p_{i},p_{j}\right)$= 1; otherwise, $Sim_{sem}\left(p_{i},p_{j}\right)$ [20,46] is equal to
(5)$$\begin{array}{c} \frac{\sum_{l_{i}\in L_{i}}\left(\max_{l_{j}\in L_{j}}{\mathrm{Sim}}_{\mathrm{sem}}\left(l_{i},l_{j}\right)\right)+\sum_{l_{j}\in L_{j}}\left(\max_{l_{i}\in L_{i}}{\mathrm{Sim}}_{\mathrm{sem}}\left(l_{j},l_{i}\right)\right)}{|L_{i}\left|+\right|L_{j}|} \end{array}$$
where $|L_{i}\left|,\right|L_{j}|$ denotes the number of semantic meanings of $L_{i}$ and $L_{j}$, respectively.

### 3.3. Extracting Similar Sub-Trajectories

PP-TPS uses the area enclosed by a real sub-trajectory and a historic sub-trajectory to measure the similarity of the two sub-trajectories. The size of the area is called the “area distance” between the two sub-trajectories. When the area distance is smaller, PP-TPS deems them to be closer. If the area distance $Area(subtr_{i},subtr_{j})$ between the two sub-trajectories $subtr_{i}$ and $subtr_{j}$ is less than a given threshold $\delta_{area}$, PP-TPS considers that they are similar sub-trajectories.
(6)$$Area\left(subtr_{i},subtr_{j}\right)<\delta_{area}$$

Note that the start points and end points of the two sub-trajectories are similar points. The area threshold $\delta_{area}$ is determined by the system.

We use the triangulation method [47] to evaluate the area enclosed by two sub-trajectories. We give an example (shown in Figure 2) to illustrate the specific evaluation process. Suppose, in Figure 2, the real sub-trajectory is $\{\left(x_{1},y_{1}\right)$, $\left(x_{2},y_{2}\right)$, …, $\left(x_{n^{\prime }},y_{n^{\prime }})\right\}$, and the historic sub-trajectory is $\{\left(a_{1},b_{1}\right)$, $\left(a_{2},b_{2}\right)$, …, $\left(a_{m^{\prime }},b_{m^{\prime }})\right\}$, where $\left(x_{1},y_{1}\right)$ = $\left(a_{1},b_{1}\right)$, $\left(x_{n},y_{n}\right)$ = $\left(a_{m},b_{m}\right)$ and $n^{\prime }$ and $m^{\prime }$ are usually small integers. First, we obtain the area of the triangle formed by the three location points $(x_{i}$, $y_{i})$, $(x_{i+1}$, $y_{i+1})$, $(a_{2}$, $b_{2})$ by calculating $S_{i}$ = $\frac{1}{2}$·$(x_{i}y_{i+1}$ + $x_{i+1}b_{2}$ + $a_{2}y_{i}$ − $x_{i}b_{2}$ − $x_{i+1}y_{i}$ − $a_{2}y_{i+1})$, where *i* is between 1 and $n^{\prime }-1$. Then, we calculate the area $S_{n}^{\prime }$ of the triangle formed by $(a_{2}$, $b_{2})$, $(a_{m^{\prime }-1}$, $b_{m^{\prime }-1})$, $(a_{m^{\prime }}$, $b_{m^{\prime }})$, the area $S_{n^{\prime }+1}$ of the triangle formed by $(a_{2}$, $b_{2})$, $(a_{m^{\prime }-2}$, $b_{m^{\prime }-2})$, $(a_{m^{\prime }-1}$, $b_{m^{\prime }-1})$, …, and the area $S_{n^{\prime }+m^{\prime }-4}$ of the triangle formed by $(a_{2}$, $b_{2})$, $(a_{3}$, $b_{3})$, $(a_{4}$, $b_{4})$. Finally, the area of the two sub-trajectories is *S* = $S_{1}$ + $S_{2}$ + … + $S_{n^{\prime }+m^{\prime }-4}$.

## 4. PP-TPS Description

The main symbols used in this article and their explanations are shown in Table 2.

### 4.1. Method Framework

The objective of PP-TPS is to establish *k*-anonymity for a provided trajectory, denoted as RT, using a given historic trajectory set HS, prior to the publication of trajectories. Figure 3 illustrates PP-TPS’s framework and workflow. PP-TPS consists of three stages: the preprocessing stage, the generating a candidate sub-trajectory stage, and the building a dummy trajectory stage.

In the preprocessing stage, PP-TPS extracts RT’s start point $p_{1}$, end point $p_{n}$, and local extremum points $p_{2}$, …, $p_{n-1}$. Subsequently, PP-TPS intercepts the part of RT from $p_{i}$ to $p_{i+1}$ to take as $subt_{i}$ and chooses the trajectories that are the same as the start and end points of RT from HS. The set of filtered trajectories is denoted as ${\mathrm{HS}}^{\prime }$.

After that, PP-TPS enters the “generating a candidate sub-trajectory stage”. In this stage, PP-TPS first finds all points on the trajectories included by ${\mathrm{HS}}^{\prime }$ that are similar to $p_{i}$ (*i*= 2, …, n−1) in the aspects of time, space, and semantics, and uses $P_{i}$ to denote the set of the found points. Note that $P_{1}$ and $P_{n}$ are set to $\left\{p_{1}\right\}$ and $\left\{p_{n}\right\}$, respectively. Subsequently, PP-TPS searches a point $sp_{i}$ from $P_{i}$ and a point $sp_{i+1}$ from $P_{i+1}$ (*i* = 1, 2, …, n−1), both of which have the same label, indicating that $sp_{i}$ and $sp_{i+1}$ come from the same historic trajectory. If PP-TPS finds such $sp_{i}$ and $sp_{i+1}$, it intercepts the segment of the corresponding historic trajectory between $sp_{i}$ and $sp_{i+1}$ and pushes this segment into ${\mathrm{SUBT}}_{i}$. Then, PP-TPS further filters the sub-trajectories of ${\mathrm{SUBT}}_{i}$ based on their enclosed areas with $subt_{i}$ to obtain the candidate sub-trajectory set ${{\mathrm{SUBT}}^{\prime }}_{i}$. Suppose the size of the space contained with one sub-trajectory of ${\mathrm{SUBT}}_{i}$ and $subt_{i}$ is less than $\delta_{area}$. In that case, PP-TPS considers that this sub-trajectory is similar to $subt_{i}$, a candidate used to create RT’s dummy trajectories.

In last stage, PP-TPS creates k−1 dummy trajectories of RT based on candidate sets ${{\mathrm{SUBT}}^{\prime }}_{1}$, ${{\mathrm{SUBT}}^{\prime }}_{2}$, …, ${{\mathrm{SUBT}}^{\prime }}_{n-1}$.

### 4.2. Preprocessing Stage

This stage mainly involves processing a given real trajectory RT and coarse screening a given historic trajectory HS. The specific operations of preprocessing include:Step 1: PP-TPS extracts the beginning point $p_{1}$ and the ending point $p_{n}$ of RT (suppose PP-TPS will extract a total of *n* location points from RT).Step 2: PP-TPS, according to $p_{1}$ and $p_{n}$, narrows down the given historic trajectory HS to obtain a preliminary candidate historic trajectory set ${\mathrm{HS}}^{\prime }$. The beginning and ending of each trajectory in ${\mathrm{HS}}^{\prime }$ are $p_{1}$ and $p_{n}$, respectively.Step 3: PP-TPS, according to Equation (Equation 1), picks out local extremum points $p_{2}$, …, $p_{n-1}$ from RT and intercepts the part of RT from $p_{i}$ to $p_{i+1}$ to take as $subt_{i}$ (*i* = 1, 2, …, n−1).

### 4.3. Generating a Candidate Sub-Trajectory Stage

The main task of this stage is to generate the candidate sub-trajectory set ${{\mathrm{SUBT}}^{\prime }}_{i}$ for $subt_{i}$ (*i* = 1, 2, …, n−1). To achieve this, PP-TPS first carries out the $AL_{POINT}$ algorithm (Algorithm 1) to obtain the point sets $P_{i}$, which include the points extracted from historic trajectories in ${\mathrm{HS}}^{\prime }$ and are similar to $p_{i}$ (*i* = 1, …, *n*) in the aspects of time, space, and semantics. Then, according to $P_{i}$, PP-TPS runs the $AL_{SUBT}$ algorithm (Algorithm 2) to generate the candidate sub-trajectory set, ${{\mathrm{SUBT}}^{\prime }}_{i}$.
**Algorithm 1** Extracting similar points $AL_{POINT}$.**Input:** ${\mathrm{HS}}^{\prime }$; $p_{1}$, $p_{2}$, …, $p_{n}$; Δt; $\delta_{spa}$, $\delta_{time}$, $\delta_{sem}$**Output:** $P_{1}$, $P_{2}$, …, $P_{n}$. 1:Let $P_{1}$←*∅*, …, $P_{n}$←*∅*; 2:Set a certain time offset Δt; 3:**for** i=1 to *n* **do** 4: **for all** H∈HS **do** 5:  **if** $|p_{H}.T-p_{i}.T|$ < Δt **then** 6:   $P_{i}$←$P_{i}$⋃$\left\{p_{H}\right\}$; 7:  **end if** 8: **end for** 9:**end for**10:**for** i=1 to *n* **do**11: **for all** $p_{i,j}$∈$P_{i}$ **do**12:  Calculate $Sim_{spa}\left(p_{i},p_{i,j}\right)$ according to Equation (Equation 3);13:  **if** $Sim_{spa}\left(p_{i},p_{i,j}\right)$ ≤ $\delta_{spa}$ **then**14:   Remove $p_{i,j}$ from $P_{i}$;15:  **end if**16: **end for**17:**end for**18:**for** i=1 to *n* **do**19: **for all** $p_{i,j}$∈$P_{i}$ **do**20:  Calculate $Sim_{time}\left(p_{i},p_{i,j}\right)$ according to Equation (Equation 4);21:  **if** $Sim_{time}\left(p_{i},p_{i,j}\right)$ ≤ $\delta_{time}$ **then**22:   Remove $p_{i,j}$ from $P_{i}$;23:  **end if**24: **end for**25:**end for**26:**for** i=1 to *n* **do**27: **for all** $p_{i,j}$∈$P_{i}$ **do**28:  Calculate $Sim_{sem}\left(p_{i},p_{i,j}\right)$ according to Equation (Equation 5);29:  **if** $Sim_{sem}\left(p_{i},p_{i,j}\right)$ ≤ $\delta_{sem}$ **then**30:   Remove $p_{i,j}$ from $P_{i}$;31:  **end if**32: **end for**33:**end for**34:**return** $P_{1}$, $P_{2}$, …, $P_{n}$;

**Algorithm 2** Extracting a candidate sub-trajectory $AL_{SUBT}$.
**Input:** $P_{1}$, $P_{2}$, …, $P_{n}$; ${\mathrm{HS}}^{\prime }$; $subt_{1}$, $subt_{2}$, …, $subt_{n-1}$
**Output:** ${{\mathrm{SUBT}}^{\prime }}_{1},{{\mathrm{SUBT}}^{\prime }}_{2},\ldots ,{{\mathrm{SUBT}}^{\prime }}_{n-1}$.
 1:Let ${{\mathrm{SUBT}}^{\prime }}_{1}$←*∅*, …, ${{\mathrm{SUBT}}^{\prime }}_{n-1}$←*∅*; 2:**for** i=1 to n−1 **do** 3: **for** j=1 to $|P_{i}|$ **do** 4:  **for** k=1 to $|P_{i+1}|$ **do** 5:   **if** $P_{i}$’s *j*th point and $P_{i+1}$’s *k*th point have same trajectory label **then** 6:    Find the historic trajectory from ${\mathrm{HS}}^{\prime }$; 7:    Intercept the segment of the trajectory between the two points; 8:    Add this segment into ${{\mathrm{SUBT}}^{\prime }}_{i}$; 9:   **end if**10:  **end for**11: **end for**12:
**end for**
13:**for** i=1 to n−1 **do**14: **for** j=1 to $|mathcal{SUBT^{\prime }}_{i}|$ **do**15:  Calculate the area *S* enclosed by the *i*th item of ${{\mathrm{SUBT}}^{\prime }}_{i}$ and $subt_{i}$ by running the $AL_{AREA}$ algorithm (Algorithm 3);16:  **if** *S* ≥ $\delta_{area}$ **then**17:   Remove the item from ${{\mathrm{SUBT}}^{\prime }}_{i}$;18:  **end if**19: **end for**20:
**end for**
21:**return** ${{\mathrm{SUBT}}^{\prime }}_{1}$, ${{\mathrm{SUBT}}^{\prime }}_{2}$, …, ${{\mathrm{SUBT}}^{\prime }}_{n-1}$;


**Algorithm 3** Calculating the Area $AL_{AREA}$.
**Input:** $seg_{1}$, $seg_{2}$
**Output:** The area *S* enclosed by $seg_{1}$ and $seg_{2}$.
  1:Let *S*← 0;  2:Take $n^{\prime }$ points ($p_{1}^{\left(1\right)}$, $p_{2}^{\left(1\right)}$, …, $p_{n^{\prime }}^{\left(1\right)}$) from $seg_{1}$;  3:Take $m^{\prime }$ points ($p_{1}^{\left(2\right)}$, $p_{2}^{\left(2\right)}$, …, $p_{m^{\prime }}^{\left(2\right)}$) from $seg_{2}$;  4:**for** i=1 to $n^{\prime }-1$ **do**  5: Calculate the area $S_{▵}$ of the triangle defined by points $p_{2}^{\left(2\right)}$, $p_{i}^{\left(1\right)}$, $p_{i+1}^{\left(1\right)}$;  6: *S*←*S*+$S_{▵}$;  7:
**end for**
  8:**for** $i=m^{\prime }-1$ to 4 **do**  9: Calculate the area $S_{▵}$ of the triangle defined by points $p_{2}^{\left(2\right)}$, $p_{i}^{\left(2\right)}$, $p_{i-1}^{\left(2\right)}$; 10: *S*←*S*+$S_{▵}$; 11:
**end for**
 12:**return***S*;


### 4.4. Building a Dummy Trajectory Stage

In this stage, based on the candidate sub-trajectory sets (${{\mathrm{SUBT}}^{\prime }}_{1}$, ${{\mathrm{SUBT}}^{\prime }}_{2}$, …, ${{\mathrm{SUBT}}^{\prime }}_{n-1}$), PP-TPS can generate k−1 dummy trajectories for the real trajectory RT by adopting various strategies, which, together with the user’s real trajectories, form an anonymized set containing *k* trajectories in order to *k*-anonymize the real trajectories, such as randomized policy, top-k−1 policy, and so on. Here, we use the randomized policy to create dummy trajectories according to similar historic sub-trajectories. PP-TPS runs the $AL_{DUMT}$ algorithm (Algorithm 4) to realize *k*-anonymity for RT.
**Algorithm 4** Building a dummy trajectory $AL_{DUMT}$.**Input:** ${{\mathrm{SUBT}}^{\prime }}_{1}$, ${{\mathrm{SUBT}}^{\prime }}_{2}$, …, ${{\mathrm{SUBT}}^{\prime }}_{n-1}$; *k***Output:** The dummy set DT of size k−1. 1:Let DT←*∅*; 2:**for** i=1 to k−1 **do** 3: **for** j=1 to n−1 **do** 4:  **if** *j* is equal to 1 **then** 5:   Randomly choose a sub-trajectory $st_{j}$ from ${{\mathrm{SUBT}}^{\prime }}_{j}$; 6:  **else** 7:   Choose a sub-trajectory $st_{j}$ from ${{\mathrm{SUBT}}^{\prime }}_{j}$ and ensure that $st_{j}$’s beginning is $st_{j-1}$’s ending; 8:  **end if** 9: **end for**10: Add the dummy trajectory $\{st_{1}$, $st_{2}$, …$st_{n-1}\}$ into DT;11:**end for**12:**return** DT;

### 4.5. Trajectory Publishing Method PP-TPS

PP-TPS adopts the *k*-anonymity mechanism to achieve the secure release of a given trajectory RT. To achieve this, PP-TPS introduces an innovative approach to establish trajectory *k*-anonymity. The $AL_{PP-TPS}$ algorithm (Algorithm 5) illustrates the entire process.
**Algorithm 5** Secure trajectory publishing $AL_{PP-TPS}$.**Input:** RT, HS**Output:** The release set PT.1:Let PT←{RT};2:Implement preprocess operations to obtain $p_{1}$, …, $p_{n}$; ${\mathrm{HS}}^{\prime }$, $subt_{1}$, …, $subt_{n-1}$;3:Set the values of Δt, $\delta_{spa}$, $\delta_{time}$, $\delta_{sem}$, $\delta_{area}$, *k*;4:Run $AL_{POINT}$($p_{1}$, …, $p_{n}$; ${\mathrm{HS}}^{\prime }$; Δt, $\delta_{spa}$, $\delta_{time}$, $\delta_{sem}$) to get $P_{1}$, $P_{2}$, …, $P_{n}$;5:Carry out $AL_{SUBT}$($P_{1}$, $P_{2}$, …, $P_{n}$; ${\mathrm{HS}}^{\prime }$; $subt_{1}$, …, $subt_{n-1}$) to obtain ${{\mathrm{SUBT}}^{\prime }}_{1}$, ${{\mathrm{SUBT}}^{\prime }}_{2}$, …, ${{\mathrm{SUBT}}^{\prime }}_{n-1}$;6:Implement $AL_{DUMT}$(${{\mathrm{SUBT}}^{\prime }}_{1}$, ${{\mathrm{SUBT}}^{\prime }}_{2}$, …, ${{\mathrm{SUBT}}^{\prime }}_{n-1}$; *k*) to get DT;7:PT←PT⋂DT;8:**return** PT;

## 5. Experiment Analysis

### 5.1. Experimental Database

The trajectory dataset used in this study was sourced from the Geolife project [48] conducted by Microsoft Research Asia. The data were collected over a period of five years from April 2007 to August 2012 by 182 users. The dataset includes user ID, trajectory dates, latitude, longitude, and the time of each location point in the GeoLife format. It comprises 17,621 trajectories primarily originating from Beijing, China. Our goal is to evaluate the efficiency of PP-TPS in achieving *k*-anonymized trajectories using this dataset. Figure 4 provides separate visual representations of the trajectory distribution for the 14th user in the Geolife dataset, both on a GPS map (Figure 4a) and within the latitude/longitude coordinate system (Figure 4b).

The GeoLife trajectory dataset does not include location names in the location points of the trajectories. To add this information, we used a process called inverse geocoding. This allows us to assign semantics and attributes to the location points. However, because the time intervals between location points are so short, multiple location points can refer to a single location. To deal with this, we selected the location points with the same location name and those that are closer in time to the midpoint for computation. We then remove the other location points. We filtered a total of 4245 trajectories from the original data, which belonged to 30 users. Location semantics were then integrated into the location points on these trajectories to facilitate the calculation of semantic similarity. Please refer to Table 3 for statistics regarding the actual dataset used.

Experiments were conducted using the PyCharm 2022 development platform, and the algorithms were implemented in Python. The experiments were conducted on a hardware environment that consisted of an Intel Core i5-13600KF 3.50 GHz processor and a Windows 11 64 GB RAM operating system. The manufacturer of the CPU is Intel Corporation, located in Santa Clara, CA, USA; the manufacturer of the operating system is Microsoft Corporation, located in Redmond, WA, USA.

### 5.2. Experiment Setup

The semantics of the experimental locations are classified into 10 types: transportation, education, work, home, industry, healthcare, food, entertainment, shopping, and outdoor. More details are shown in Table 1. Table 4 displays the specific experiment parameters. The anonymity level *k* ranges from 3 to 15.

We chose three classical methods to compare with PP-TPS:(1)Random scheme [49]: The Random method randomly selects k−1 locations from the candidate locations to generate dummy trajectories.(2)MTPPA scheme [50]: The MTPPA method filters the historic trajectories based on the user’s stopping point and movement rate. Subsequently, it selects k−1 historic trajectories to compose a *k*-anonymous set.(3)LRM scheme [7]: The LRM method measures the probabilistic and geographic similarity between pairs of locations and utilizes this information to select sampled historic locations. These locations are then categorized into different equivalent probability groups. Afterwards, the scheme merges sampled locations from each equivalent probability class to create a trajectory.

### 5.3. Experimental Results

#### 5.3.1. Anonymous Success Rate Analysis

In the context of privacy protection within location-based services, a higher level of anonymity ensures that a user’s query location is more effectively concealed. This makes it harder for attackers to figure out the user’s coordinates, resulting in better privacy protection. This translates to a higher degree of privacy protection. Conversely, a lower degree of anonymity implies a lower level of privacy protection. In this paper, we analyze the privacy protection degree using the anonymity success rate, depicted in Figure 5. The figure illustrates the anonymity success rates for various numbers of mobile users across varying anonymity privacy degrees.

The anonymity success rate is a crucial metric for evaluating the effectiveness of location anonymity algorithms. A higher anonymity success rate corresponds to a more effective anonymity algorithm. Formally expressed as SR = $\left|\frac{S^{\prime }}{S}\right|$, the anonymization success rate represents the percentage of messages that are successfully anonymized out of all anonymization requests made by mobile users.

The vertical coordinate (anonymization success rate) and the horizontal coordinate (anonymity degree *k*) in Figure 5 exhibit the anonymization success rates of several methods under various anonymity degrees. The Random method exhibits a notably low anonymization success rate due to the easily recognizable randomly generated trajectories, resulting in the lowest success rate. MTPPA has a mediocre success rate in achieving anonymity. Attackers can recognize actual trajectories through temporal reachability, since this approach does not limit the time range of the anonymous trajectories. Although LRM takes into account more dimensions than MTPPA, the possibility of semantic leaks remains. Our PP-TPS, in contrast, guarantees that each dummy trajectory cannot be identified based on multidimensional similarity, leading to a better anonymization success rate. It also places restrictions on the area enclosed by the trajectories. Overall, the average anonymization success rate increases with higher levels of anonymity (*k*), as increasing the anonymized trajectories makes it increasingly challenging for attackers to identify real trajectories.

#### 5.3.2. Relative Anonymity Analysis

Relative anonymity is a measure of the proportion between the covered locations within the anonymous area and the anonymity level *k*, and this proportion must never be less than 1. In other words, the greater the relative anonymity, the greater the number of users involved, indicating a higher level of privacy protection. Figure 6 displays the trend of relative anonymity with respect to the change in *k* value. The horizontal axis represents the anonymity level *k*, while the vertical axis represents the relative anonymity.

The objective of the Random approach is to randomly choose locations by generating dummy trajectories. Since each location has an equal probability of selection, they all have the same level of recognizability. According to the definition of relative anonymity, when *k* is fixed, the larger the total number of locations, the greater the relative anonymity. Conversely, when the total number of locations is fixed, increasing the anonymity parameter *k* decreases relative anonymity. Although the randomized algorithm selects locations randomly, resulting in an unstable size of the anonymity region, the overall trend (average value) in relative anonymity is still relatively low. MTPPA and LRM utilize historic trajectory information to achieve more accurate anonymization. They rely on trajectory similarity or positional similarity, resulting in the formation of larger anonymized regions and a higher relative anonymity compared to the Random scheme. Our PP-TPS method, in comparison to the others, not only focuses on the similarity of location points but also filters historic sub-trajectories by the area enclosed by the sub-trajectories, ensuring a better accomplishment of *k*-anonymity and aligning with the presented experimental results.

#### 5.3.3. Data Availability Analysis

In this paper, we use the available data rate (AD) to evaluate the number of available data in the anonymized set, which is computed as follows: AD = $\frac{num\_reach(k_{set})}{num\_all(k_{set})}$, denoting the number of reachable trajectories in the *k*-anonymity set over the number of available trajectories in the upper *k*-anonymity set.

Figure 7 assesses the available data rate across the four approaches. It can be observed that the data availability rate is 1 for all approaches except the Random method, which varies depending on the trajectory anonymity level *k*. This discrepancy is because the other schemes rely on historic trajectories and generate real trajectories using different algorithms. It is important to note that the data availability rate for the Random method falls below 1. This is attributed to the random selection of locations for real trajectory generation in this scheme, which may include points in inaccessible areas, such as the center of a large river, where location points cannot be generated. Our PP-TPS approach filters location points based on historic track protection, which are real user-generated location points containing real information, all of which are available data.

#### 5.3.4. Location Entropy Analysis

The effectiveness of positional privacy protection in trajectory semantics can be evaluated by using location entropy [51], which measures the effectiveness of positional privacy protection. When positional entropy is high, the trajectory semantics become more alike. The formula used to calculate positional entropy is as follows: $H_{p}=-\frac{1}{N}\sum_{i=1}^{N}log_{2}\left(\frac{n_{i}}{T}\right)$. Among these variables, *N* represents the number of trajectory positions, $n_{i}$ represents the frequency of semantics at position *i*, and *T* represents the total frequency of all semantics in the text. Figure 8 displays the experimental comparison of various anonymity levels.

As the anonymity level increases, the position entropy of the three algorithms increases, and position privacy protection intensifies. Our algorithm takes into account the semantic similarity of trajectories, making it impossible for attackers to identify the true trajectory through semantics. This results in a significantly higher position entropy compared to the other two algorithms, making our algorithm superior. Among these methods, MTPPA employs historic trajectories as virtual trajectories, increasing the likelihood of matching the semantics of actual trajectories compared to random algorithms. Consequently, MTPPA also exhibits a higher position entropy than random algorithms.

#### 5.3.5. Anonymous Location Runtime Analysis

Moreover, we compare the running times of Random, MTPPA, LRM, and PP-TPS, also called the anonymous trajectory generation time. This is because the other operational steps involved in the three schemes are largely similar and require similar times. The experiments compare the anonymous trajectory generation times of Random, MTPPA, LRM, and PP-TPS. The horizontal coordinate represents the number of anonymous trajectories, denoted as the degree of anonymity *k*. It is clear that when *k* rises, the time required by the three strategies grows in distinct ways. Random takes the shortest time, since it directly and randomly produces anonymous trajectories. PP-TPS requires additional processing for calculating the similarity of location points and the area enclosed by the sub-trajectory. However, it primarily operates on location points and traverses the length of the real trajectory only twice to calculate the sub-trajectory area. Therefore, it has a comparatively shorter running time compared to LRM and MTPPA. These findings indicate that PP-TPS can reduce computational effort while enhancing trajectory privacy. Figure 9 represents the generation time for anonymized locations across different values of *k*.

## 6. Conclusions

This paper proposes an efficient method, PP-TPS, to realize the *k*-anonymity of a trajectory. PP-TPS involves filtering historic location points by calculating spatial, temporal, and semantic similarities. Based on the discovered similar points, PP-TPS can extract a large amount of corresponding sub-trajectories from many historic trajectories. Then, PP-TPS sifts through these historic sub-trajectories further according to the area distance between them and uses the sub-trajectories of the real trajectory to obtain candidate sub-trajectory sets. Finally, PP-TPS generates k−1 dummy trajectories for the real trajectory based on the candidate sub-trajectory sets.

PP-TPS guarantees that the generated dummy trajectory is similar to the real trajectory at the point level by carefully considering the multidimensional similarity of the location points. Furthermore, it measures the similarity between sub-trajectories using the area distance. This novel approach provides a more straightforward and efficient way to handle data when compared to computing the similarity of the entire trajectory, making the solution more practical and effective. Overall, PP-TPS surpasses traditional privacy methods, reduces computing costs, and achieves a satisfactory balance. Additionally, the method introduces a novel metric for trajectory similarity based on area. Moreover, PP-TPS utilizes real historic sub-trajectories to generate dummy trajectories, making them more authentic and providing better privacy protection for real trajectories, Although our approach has achieved some success in protecting user privacy, there are still some areas for improvement. Specifically, the challenges we faced included the sparsity of user historic trajectories, which made it relatively difficult to generate a sufficient number of virtual trajectories in some scenarios; at the same time, we also need to address the challenge of meeting the individual needs of different users, which requires us to further optimize the model to be more flexible in meeting user expectations in different scenarios.

In future work, we plan to extend the functionality of the system to provide enhanced privacy protection. This includes work in the following areas: (1) We will think about the possibility of filtering trajectories by sub-trajectory similarity in more dimensions and explore different distance computation methods in multiple dimensions to investigate more practical privacy-preserving solutions for trajectories. (2) Our emphasis will be on exploring the privacy-preserving impact of the algorithm when dealing with sparsely populated historic trajectories. Our objective is to tackle the challenges presented by sparse data to guarantee robust privacy preservation. (3) Our upcoming endeavors involve the creation of a personalized trajectory privacy protection model. This model will empower users to autonomously define the anonymization region for location points and determine the level of trajectory anonymization based on their unique privacy requirements. This approach aims to cater to the varied needs of distinct users and diverse scenarios.

## Figures and Tables

**Figure 1 sensors-23-09652-f001:**
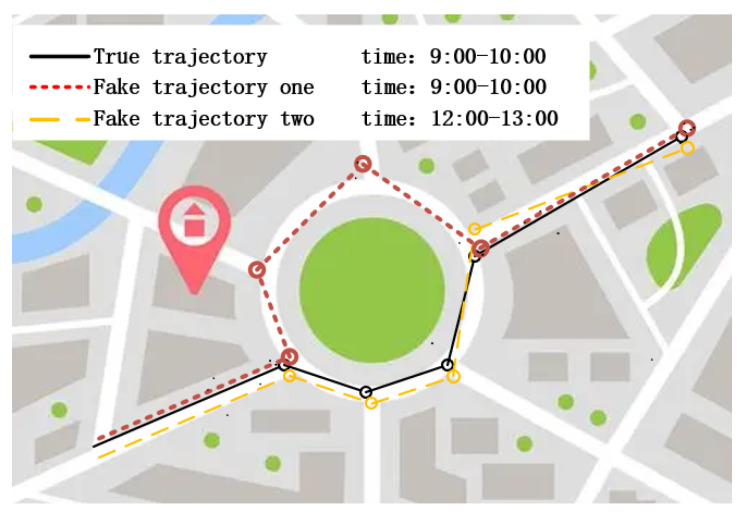
Example of dummy trajectory identification.

**Figure 2 sensors-23-09652-f002:**
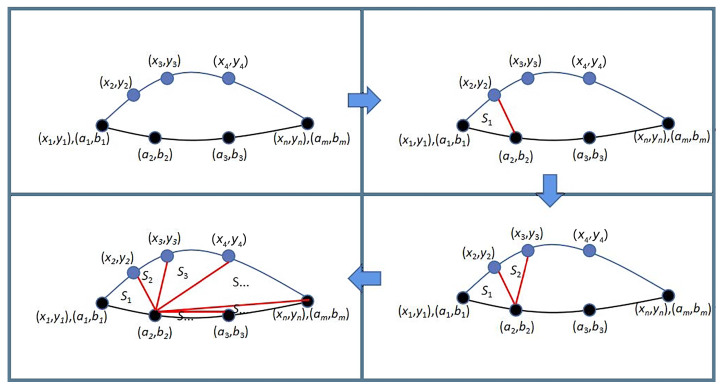
Example of evaluating area size.

**Figure 3 sensors-23-09652-f003:**
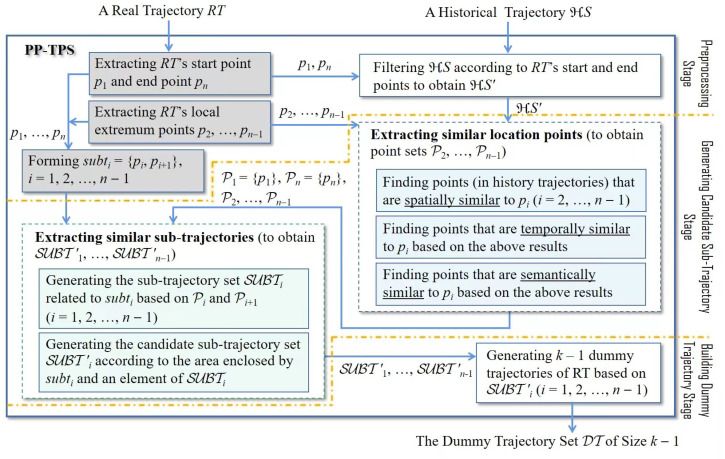
Overall flowchart of PP-TPS.

**Figure 4 sensors-23-09652-f004:**
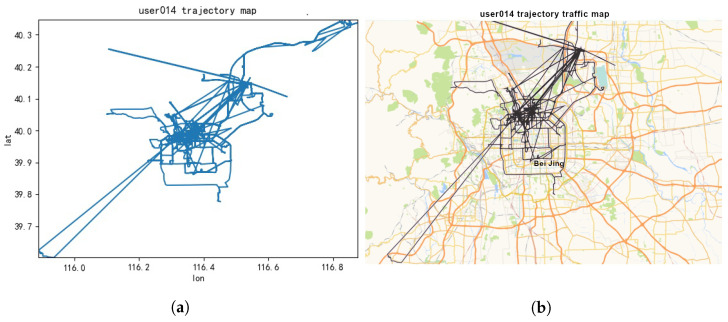
Trajectory map of the 14th user. (**a**) GPS map; (**b**) latitude and longitude coordinate system.

**Figure 5 sensors-23-09652-f005:**
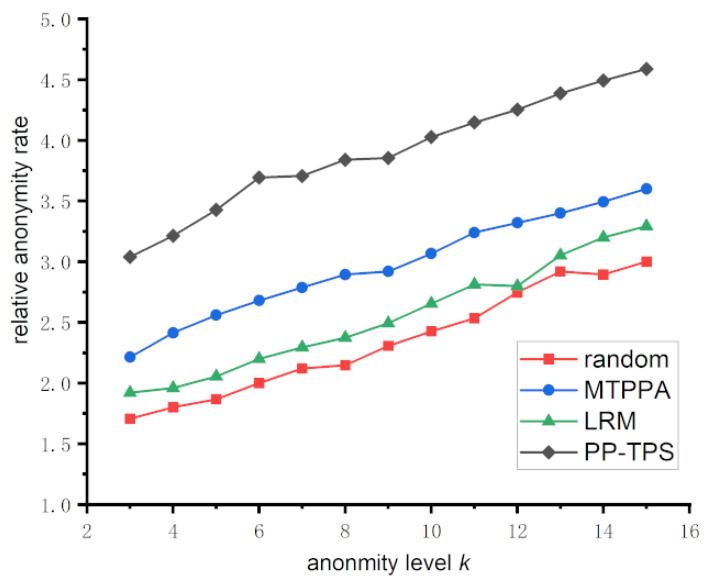
Average anonymity success rate of different anonymity degrees.

**Figure 6 sensors-23-09652-f006:**
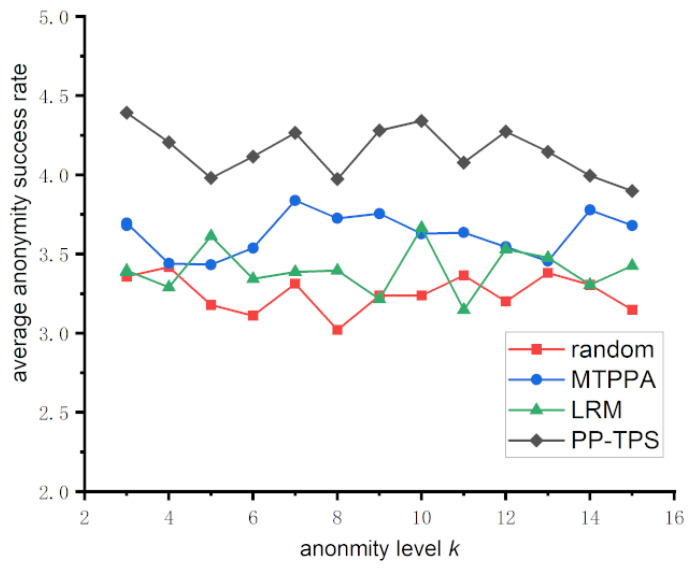
Relative anonymity under different anonymity degrees.

**Figure 7 sensors-23-09652-f007:**
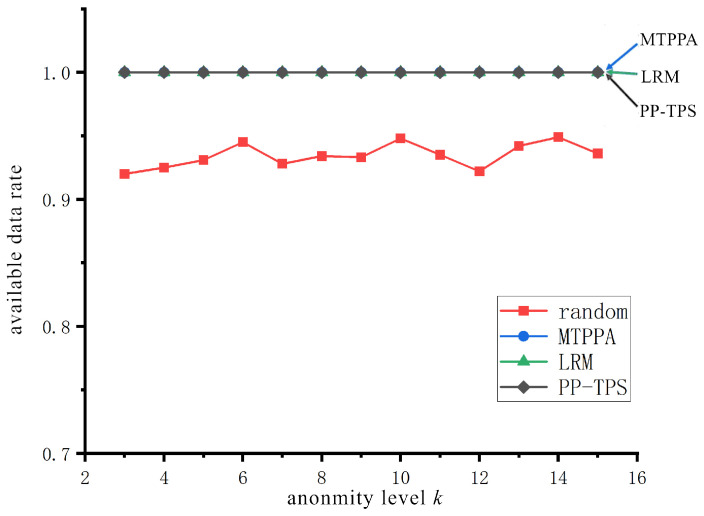
Available data rates at different anonymity levels.

**Figure 8 sensors-23-09652-f008:**
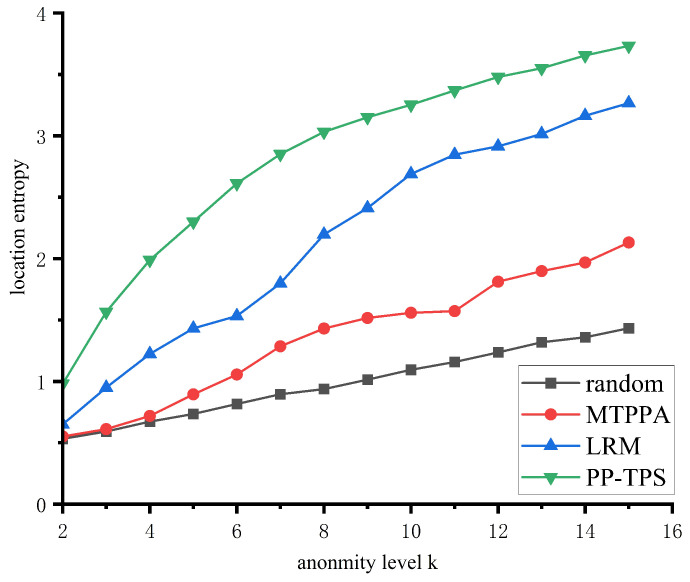
Location entropy at different anonymity levels.

**Figure 9 sensors-23-09652-f009:**
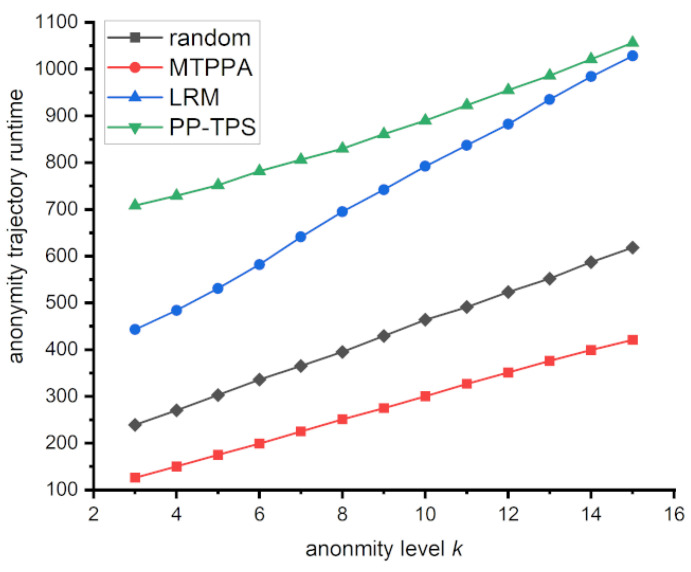
Anonymous Location Runtime at different anonymity levels.

**Table 1 sensors-23-09652-t001:** Location semantics categories.

Semantic Category	Detailed Information
Transportation	Airport, railway, bus station
Education	Campus, training center
Work	Government, office, building, company
Home	Apartment, house
Industry	Factory, industrial estate, etc.
Health care	Hospital, medical center, pharmacy
Food	Restaurant
Entertainment	Hotel, gym, leisure
Shopping	Shop, mall, outlet
Outdoor	Park, sports field

**Table 2 sensors-23-09652-t002:** Main notation.

Symbol	Explanation
RT = $\left\{p_{1},\ldots ,p_{n}\right\}$	A real trajectory
HS	A historic trajectory set
${\mathrm{HS}}^{\prime }$	The historic trajectory set obtained after initially screening HS
H	A historic trajectory in ${\mathrm{HS}}^{\prime }$
$p_{H}$	A location point on H
$P_{i}$	A point set related to $p_{i}$ (*i* = 1, 2, …, *n*)
$subt_{i}$ = $\{p_{i}$, $p_{i+1}\}$	The sub-trajectory between points $p_{i}$ and $p_{i+1}$ of RT (*i* = 1, 2, …, n−1)
${{\mathrm{SUBT}}^{\prime }}_{i}$	The candidate sub-trajectory set filtered out from ${\mathrm{SUBT}}_{i}$
DT	The dummy trajectory set of size k−1 for RT
$\delta_{spa}$, $\delta_{time}$	The spatial threshold and the temporal threshold
$\delta_{sem}$, $\delta_{area}$	The semantic threshold and the area threshold
$p_{H}.T$	The time attribute of point $p_{H}$
$p_{i}.T$	The time attribute of point $p_{i}$
$p_{i,j}$	The *j*th location point of $P_{i}$

(1) $p_{1}$ is RT’s start point, $p_{n}$ is RT’s end point, and $p_{i}$ (*i*= 2, 3, …, n−1) is RT’s local extremum point. (2) We suppose that every trajectory of HS has a unique label. (3) Every trajectory belonging to ${\mathrm{HS}}^{\prime }$ has the same start and end points as RT. (4) Every point of $P_{i}$ (extracted from ${\mathrm{HS}}^{\prime }$) is similar to $p_{i}$ in three dimensions (such as space, time, semantics). Each node in set $P_{i}$ carries a label that is the label of the historical trajectory from which it came. $P_{1}$ (respectively, $P_{n}$) includes only one point $p_{1}$ (respectively, $p_{n}$) with a special label which means that the point comes from all trajectories in ${\mathrm{HS}}^{\prime }$. (5) $sp_{i}$∈$P_{i}$, $sp_{i+1}$∈$P_{i+1}$, and the label attached to $sp_{i}$ are the same as $sp_{i+1}$. Therefore, $\{sp_{i}$, $sp_{i+1}\}$ is a part of the historic trajectory (∈${\mathrm{HS}}^{\prime }$) labeled by this label. (6) If the area closed by a sub-trajectory belonging to ${\mathrm{SUBT}}_{i}$ and $subt_{i}$ is less than the threshold $\delta_{area}$, then the sub-trajectory is a member of ${{\mathrm{SUBT}}^{\prime }}_{i}$.

**Table 3 sensors-23-09652-t003:** Geolife statistics data.

Geolife Statistics Data	Value
Number of trajectories	4242
Average trajectory length	1662.61
Number of location points	7,052,782
Average length of time per trajectory (hours)	3.78

**Table 4 sensors-23-09652-t004:** Experimental parameters.

Parameter	Detail
$\delta_{spa}$/$\delta_{time}$/$\delta_{sem}$	0.5/0.7/0.5
$\delta_{area}$/*k*	0.5/3–15
Location semantics	Transportation, Education, Work, Home, Industry, Healthcare, Food, Entertainment, Shopping, Outdoor

## Data Availability

Data are contained within the article.

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
