# Peer review of "A Privacy-Preserving Trajectory Publishing Method Based on Multi-Dimensional Sub-Trajectory Similarities"

_sensors, 2023, doi:10.3390/s23249652_

Round 1

Reviewer 1 Report

Comments and Suggestions for Authors

In this paper, authors propose a k-anonymity trajectory privacy protection method based on the similarity of sub-trajectories. The proposed method takes into account both the similarity of points and the similarity of the sub-trajectories between historical and real trajectories.

Paper is clearly written, logically organized, very comprehensive and impressive amount of research and study has been carried out. The content is technically sound and contains sufficient interest to merit publication.

The reviewer has minor comments that need to be addressed by the authors:

·       In the abstract, the result of this work must be described briefly.

·       The quality of the figures should be improved.

·      In order to highlight the innovation of this work, it is better to cite more up-to-date references.

·  Please improve the reference format and verify the number of each reference cited in the paper.

Comments on the Quality of English Language

    Some sentences are too long to make readers confused, and there are also some typos and grammar errors in this paper.

Reviewer 2 Report

Comments and Suggestions for Authors

This article addresses a crucial issue in the era of location-based services and trajectory data by proposing a k-anonymity trajectory privacy protection method that focuses on the similarity of sub-trajectories. The paper critiques existing trajectory k-anonymity methods for primarily considering point similarity, resulting in a large dummy trajectory space and significant performance overhead during trajectory selection. The proposed method offers a novel approach by incorporating both point and sub-trajectory similarities, aiming to mitigate these challenges.

The strength of this paper lies in its holistic consideration of trajectory privacy, going beyond traditional point-based approaches. By examining the similarity of sub-trajectories, the authors effectively reduce the space needed for dummy trajectories and alleviate the computational burden associated with trajectory similarity evaluation. The utilization of real historical sub-trajectories for generating dummy trajectories is a commendable addition, contributing to the authenticity of the privacy protection mechanism.

There are a few suggestions that could help improve the content and clarity of the research:

1.               Due to the background of the letters in the caption of Figure 1, the image becomes blurred. It is desirable to change this.

2.               It would be helpful to discuss the shortcomings and limitations of your model.

Reviewer 3 Report

Comments and Suggestions for Authors

after i  read this paper i want ask about some q :

1- why using : to choose suitable k − 1 dummy trajectories for a given real

2-trajectories between historical and real trajectories: can more detail

3-need more testing in ", our 15 method has better privacy protection effect, higher data quality, and better performance"

4- you begin in " purposes like traffic flow optimization [6] and" reference 6 why 1 as begin

5-in line 83" e similarity of semantic trajectories " I think more no similarity"

6-"tory of the 116 trajectory according to the computed multi-dimensional similarities between them. The 117 main contributions of this paper are as follows: " more explain the novelty 

7-in line 301"when we calculate the location  semantic similarity between pi and pj: why not pj+1

8-Figure 2. Example of Evaluating Area’s Size: can draw in another style

9-in line 376: can generate k − 1 dummy trajectories for the real trajectory, you generate K only

10-The location semantics of the experiment are categorized into 10 types:::: or 9 types?????

11-rewrite the statement:PP-TPS considers both the similarity of points and the similarity of the sub-trajectories 514 between historical real trajectories. This method reduces the number of dummy trajectories created by just considering point similarity and avoids the need to solve for complete trajectory similarity. Moreover" to focus more on novelty for that

12- the references are few we need more from mdpi journals 

Comments on the Quality of English Language

i need to see the enhancement afer do all notes
